# Radiogenomics Map-Based Molecular and Imaging Phenotypical Characterization in Localised Prostate Cancer Using Pre-Biopsy Biparametric MR Imaging

**DOI:** 10.3390/ijms25105379

**Published:** 2024-05-15

**Authors:** Chidozie N. Ogbonnaya, Basim S. O. Alsaedi, Abeer J. Alhussaini, Robert Hislop, Norman Pratt, J. Douglas Steele, Neil Kernohan, Ghulam Nabi

**Affiliations:** 1Division of Imaging Science and Technology, School of Medicine, University of Dundee, Dundee DD1 4HN, UK; chidozie.ogbonnaya@yahoo.co.uk (C.N.O.); a.j.a.h.m.alhussaini@dundee.ac.uk (A.J.A.); d.steele@dundee.ac.uk (J.D.S.); 2Statistics Department, University of Tabuk, Tabuk 47512, Saudi Arabia; balsaedi@ut.edu.sa; 3Cytogenetic, Human Genetics Unit, NHS Tayside, Ninewells Hospital and Medical School, Dundee DD1 9SY, UK; robert.hislop@nhs.scot (R.H.); norman.pratt@nhs.scot (N.P.); 4Department of Pathology, NHS Tayside, Ninewells Hospital and Medical School, Dundee DD1 9SY, UK; neil.kernohan@nhs.scot

**Keywords:** radiogenomics, biparametric magnetic resonance imaging, copy number variation

## Abstract

To create a radiogenomics map and evaluate the correlation between molecular and imaging phenotypes in localized prostate cancer (PCa), using radical prostatectomy histopathology as a reference standard. Radiomic features were extracted from T2-weighted (T2WI) and Apparent Diffusion Coefficient (ADC) images of clinically localized PCa patients (n = 15) across different Gleason score-based risk categories. DNA extraction was performed on formalin-fixed, paraffin-embedded (FFPE) samples. Gene expression analysis of androgen receptor expression, apoptosis, and hypoxia was conducted using the Chromosome Analysis Suite (ChAS) application and OSCHIP files. The relationship between gene expression alterations and textural features was assessed using Pearson’s correlation analysis. Receiver operating characteristic (ROC) analysis was utilized to evaluate the predictive accuracy of the model. A significant correlation was observed between radiomic texture features and copy number variation (CNV) of genes associated with apoptosis, hypoxia, and androgen receptor (*p*-value ≤ 0.05). The identified radiomic features, including Sum Entropy ADC, Inverse Difference ADC, Sum Variance T2WI, Entropy T2WI, Difference Variance T2WI, and Angular Secondary Moment T2WI, exhibited potential for predicting cancer grade and biological processes such as apoptosis and hypoxia. Incorporating radiomics and genomics into a prediction model significantly improved the prediction of prostate cancer grade (clinically significant prostate cancer), yielding an AUC of 0.95. Radiomic texture features significantly correlate with genotypes for apoptosis, hypoxia, and androgen receptor expression in localised prostate cancer. Integration of these into the prediction model improved prediction accuracy of clinically significant prostate cancer.

## 1. Introduction

Pre-biopsy MR imaging, especially multi-parametric sequences (e.g., T1WI, T2WI, DCE), has become the standard of care for PCa diagnosis and characterisation. MRI visible lesions are categorized using structured the Prostate Imaging Reporting and Data System (PI-RADS v2), which predicts the likelihood of clinically significant PCa [1]. However, PI-RADS v2 based interpretation of the images is limited by inter-observer variation between radiologists [2]. Reports have been made regarding the use of PI-RADS scores to predict the GS of PCa [2,3,4,5]. Additionally, recent studies have explored the potential correlations of texture features based on grey level co-occurrence matrices (GLCMs) of T2WI images and pathological differences in PCa [6,7,8]. Radiomics analysis using MR images has been reported for assessing heterogeneity in various cancers, including glioblastoma (GBM) [9,10], lung [11,12], colorectal [13,14,15] and PCa [16,17,18,19,20,21], among others. A similar approach in prostate cancer is in its early stages. Certainly, an association between genomic alteration and textural features in prostate cancer has not been reported.

In addition to radiomics features, genetic alterations in cancers determine tumor aggressiveness and growth pattern. The integration of genomics and radiomics analyses is called radiogenomics and the map generated by integration of features from both could serve as biomarker of the disease. To use radiogenomics as cancer biomarkers, features must reflect underlying pathophysiology and be reproducible. Key drivers of cancer progression, such as angiogenesis, apoptosis, and hypoxia, are known [22,23,24,25,26], but their interpretation via non-invasive radiomics analysis for PCa remains unexplored. Molecular sequencing and advanced computing enable large-scale characterization and high-throughput data extraction from MRI images [20,27,28]. These have paved the way for precision and personalized medicine. Radiogenomics approaches provide evidence for imaging biomarkers as surrogates for molecular features, facilitating molecular subtyping and insights into PCa progression and treatment response.

In the present study, the radiogenomics maps were created using pre-biopsy biparametric MRI (bpMRI) in men with clinically localized PCa. The maps linked textural features from biparametric MR images and gene expression profiles generated by DNA sequencing for patients with localized PCa. We believe this approach contributes to several new areas of knowledge. Firstly, several textural features were extracted in three different risk groups of PCa, with radical prostatectomy as the reference standard. Secondly, DNA extraction and genomic analysis were performed on matched areas of cancer between images and histology, facilitated by 3-D printed moulds as an orientation method. Finally, the significance of the radiogenomics approach was analyzed statistically to explore its potential in a predictive model aimed at Gleason score prediction.

## 2. Results

### 2.1. Patients Summary and Workflow

The distribution of demographic data of patients included in this study is shown in Table 1. A total of 15 patients with PCa were included in the prospective data collection (Figure 1). They were divided into three groups after the reclassification using the ISUP grading system [29,30]. Group 1 consisted of Gleason scores of 3 + 3; Group 2 consisted of Gleason scores of 3 + 4; and Group 3 consisted of Gleason scores of or more than 4 + 3 or 4 + 4. The workflow of this study is described in Figure 2, which shows a two-stage pipeline. Radiomic feature extraction from T2WI and ADC images occurred in stage one, followed by tissue processing, staining, DNA extraction, and correlation analysis in stage two. Results are displayed in a heatmap, and the radiogenomic model’s accuracy in predicting clinically significant prostate PCa is represented by an ROC plot.

### 2.2. Radiomic Texture Feature Analysis

The radiomic texture features were evaluated for variability using one-way ANOVA and Kruskal–Wallis tests. The significant features (*p*-value ≤ 0.05) were further investigated (Table 2) to ensure reliable and accurate results.

### 2.3. Chromosomal Analysis

The karyoview was chosen from the ChAS tool option, which displays all CNVs of the entire genome of the tested tissue samples, with red bars indicating loss and blue bars indicating gain. Typically, the size of each bar corresponds to the size of the CNV. Furthermore, using the segments tab (rather than the default Karyoview), with an emphasis on copy number gain or loss, ChAS enabled us to visualize and summarize chromosomal abnormalities throughout the genome on the Apoptotic, Androgen regulated, and hypoxia related genes, which were of particular interest.

### 2.4. Correlation Analysis

Figure 3 shows the correlation using heatmaps with Hierarchical Clustering of hypoxia genes against significant radiomic features in relation to Gleason Grade. In high grade group Prolyl 4-hydroxylase, alpha polypeptide 1 (P4HA1) significantly correlated with Angular Second Moment T2WI, Correlation T2WI, Sum Average T2WI and Sum Square Variance T2WI (r = 0.667, 0.704, 0.625 and 0.575, respectively). Angiopoietin-like protein 4 (ANGPLT4) correlated with Sum square variance T2WI, and Vascular endothelial growth factor (VEGFA) correlated with Contrast T2WI (r = 0.518 and 0.736, respectively). In the intermediate group, ANGPTL4 correlates with Angular Second Moment (r = 0.613). In the low-grade group, P4HA1 correlates with Sum square variance T2WI and Inverse difference T2WI (r = 0.702 and 0.606, respectively). VEGFA correlated with Contrast T2WI, and Sum average T2WI in high grade and low-grade cancers, respectively (0.517 and 0.606, respectively).

Figure 4 shows the correlation heatmap with Hierarchical Clustering of androgen receptor genes against significant radiomic features in relation to Gleason Grade. In the high grade, Kallikreins 2 (KLK2) correlates with Contrast T2WI (r = 0.704). In the intermediated grade, NK3 Homeobox 1 (NKX3.1) correlated with ContrastT2WI, Correlation T2WI, Sum Square VarianceT2WI and Inverse Difference T2WI (r = 0.527, 0.539, 0.636 and 0.857, respectively). KLK2 correlated with Angular Second Moment (r = 0.539). In low grade disease, NKX3.1 correlated with Angular Second Moment, Sum Square Variance T2WI and Inverse Difference T2WI (r = 0.772,0.636 and 0.857, respectively). KLK3 correlated with Sum Square Variance T2WI and Sum Average T2WI (r = 0.628 and 0.639).

Figure 5 shows the correlation heatmaps with Hierarchical Clustering of apoptosis genes against significant radiomic features in relation to Gleason Grade. In high grade, Tumor Protein p53 Binding Protein 2 (TP53BP2) correlated with Sum Square Variance T2WI, Inverse Difference T2W and Sum Average T2WI (r = 0.609, 0.652, 0.822, respectively). In intermediate grade tumors, BCL2-Associated Athanogene 3 (BAG3) correlated with Inverse difference T2WI (r = 0.604), and Tumor Necrosis Factor-Related Apoptosis-Inducing Ligand receptor 2 (TRAIL.R2) correlated with Contrast T2WI (r = 0.659). In Low grade tumors, BAG3 correlated with Contrast T2WI (r = 0.523).

### 2.5. Predictive Analysis

The combination of radiomics and genomics had a notable impact on prostate cancer Gleason score prediction. The integrated model’s accuracy was assessed using Receiver Operating Characteristic (ROC) analysis, showing a strong ability to predict clinically significant PCa (Figure 6).

## 3. Discussion

In the present study, we correlated textural image features of pre-biopsy MRI of PCa with the DNA-based arrays data to delineate radiogenomics biomarkers of localized PCa in three risk categories based on histopathological Gleason grading. The findings of this study, to the best of our knowledge and for the first time, showed correlations between key PCa pathways (apoptosis genes, hypoxia driven genetic changes and androgen receptor related genes) and radiomics textural features using pre-biopsy bpMRI scans obtained in biopsy naive men. The study used oncoScanTM FFPE kit assays for genetic analysis utilizing archived radical prostatectomy tissues. The advantages of this approach, based on molecular inversion probe technology, are well described in previous reports [31,32]. This technique allows assessment of PCa archived tissues (frozen repository of prostate cancer tissue is challenging). Correlating imaging features with molecular signatures for non-invasive imaging-based insights into cancer behavior and treatment responses, in high-grade cancers, loss of androgen receptor genes (NKX 3.1) may correlate with specific radiomics features on radiogenomics mapping, as shown in Figure 4. The map holds potential as a marker for aggressive prostate cancer and therapy, considering molecular processes and the tumor microenvironment’s influence on treatment strategies and disease prognosis.

The study showed that hypoxia related genes and the high-grade group of PCa had positive correlations as shown between P4HA1 gene and Angular Second Moment T2WI, Correlation T2WI, Sum Average T2WI, and Sum Square Variance T2WI, with coefficients ranging from 0.575 to 0.704, as shown in Figure 3. The overexpression of P4HA1, a gene encoding a collagen synthesis enzyme, has been associated with increased invasion and metastasis in various malignancies, including prostate cancer. Collagen and stromal tissues play essential roles in the tumor microenvironment, influencing tumor progression and metastasis. Studies have linked the onset and progression of prostate cancer to three genes related to hypoxia: P4HA1, ANGPLT4, and VEGFA. P4HA1 expression positively correlates with tumor stage and grade, with higher levels observed in prostate cancer tissue compared to healthy tissue [33,34]. The development and progression of prostate cancer through the P4HA1 pathway involves facilitating the growth and survival of cancer cells, activating certain cellular signaling pathways, and controlling the activity of a key transcription factor HIF-1α. By focusing on inhibiting P4HA1 and the pathways it influences, there may be potential for a viable approach to treating PCa. Contrast T2WI evaluates the variability in intensity differences, offering insights into the level of sharpness and sudden changes in texture present within the image [35]. ANGPTL4 is a member of the angiopoietin family and over expressed in cancer tissue due to hypoxia [36]. In the present study, ANGPTL4 expression was also found to correlate with Sum square variance T2WI. ANGPTL4 plays a role in angiogenesis, the process of blood vessel development, and its increased expression has been observed in several malignancies, including prostate cancer. Higher ANGPTL4 levels in prostate cancer tissue have been associated with worse overall survival rates. The active PI3K/Akt pathway induces ANGPTL4 production in response to hypoxia, promoting cancer growth [23]. VEGFA is responsible for stimulating angiogenesis and is known to be overactive in various cancers, including prostate cancer. Prostate cancer tissue demonstrated significantly higher levels of VEGFA expression compared to healthy tissue. Upregulation of VEGFA was associated with more advanced stages and grades of tumors [37].

In intermediate-grade cancer, NKX3.1 (Androgen receptor gene) showed positive correlations (ranging from 0.527 to 0.857) with Contrast T2WI, Correlation T2WI, Sum Square Variance T2WI, and Inverse Difference T2WI. KLK2 correlated with Angular Second Moment (r = 0.539). In low-grade cancer, NKX3 displayed strong positive correlations (ranging from 0.636 to 0.857) with Angular Second Moment, Sum Square Variance T2WI, and Inverse Difference T2WI. KLK3 correlated with Sum Square Variance and Sum Average (r = 0.628 and 0.639). The androgen receptor (AR) plays a vital role in prostate cancer growth and progression [38,39]. Genes regulated by AR, such as KLK2, NKX3.1, and KLK3 (PSA), are extensively studied in prostate cancer [40,41,42]. KLK2, regulated by AR, is a biomarker for PCa [42]. Reduced NKX3.1 levels are associated with aggressive PCa and disease progression [43]. KLK3 (PSA), regulated by AR, is used clinically as a biomarker for PCa progression. Monitoring changes in KLK3 levels helps track disease advancement [42].

The results for apoptotic genes revealed significant findings in high-grade PCa, where TP53BP2 exhibited positive correlations (ranging from 0.609 to 0.822) with Sum Square Variance T2WI, Inverse Difference T2WI, and Sum Average T2WI. In intermediate-grade cancer, BAG3 had a positive correlation (r = 0.604) with Inverse Difference T2WI, and TRAIL.R2 showed a positive correlation (r = 0.659) with Contrast T2WI. In low-grade cancer, BAG3 displayed a positive correlation (r = 0.523) with Contrast T2WI. TRAIL.R2 and TP53BP2 may play roles in PCa, with TRAIL.R2 promoting apoptosis and TP53BP2 regulating p53’s activity, influencing tumor growth and apoptosis resistance. However, the exact mechanisms require further investigation [44,45,46].

Textural features of images assess tissue homogeneity. This is measured via Inverse Difference. The latter is derived from the GLCM and represents the average variation in pixel intensity levels between adjacent pairs in the image. Inverse difference values range from greater to lower, with a higher value denoting a more homogeneous texture and a lower value denoting a more heterogeneous texture [47,48,49,50]. GLCM Inverse Difference T2WI can differentiate between cancerous and healthy tissue, as cancerous tissue has a more uneven texture and lower inverse difference value [48,49,50,51]. GLCM Sum of Squares Variance (SSV) is useful for evaluating prostate gland homogeneity for PCa detection in T2WI, where a higher SSV score indicates malignancy [50,52,53]. GLCM Sum Average feature analyzes T2WI texture, identifying PCa lesions as regions with lower signal strength due to malignant cells [50,54]. GLCM contrast measures grey-level variation, enabling PCa and normal tissue differentiation [50,55,56]. GLCM Correlation analysis assesses textural disparities, aiding PCa detection by measuring spatial correlation of grey-level values [50,57,58].

The proposed radiogenomics model demonstrated high predictive accuracy, achieving an AUC of 0.95 (CI: 0.88–1) and an accuracy of 0.93.

The main limitation of this study is its reliance solely on data from a single institution, coupled with the inability to perform external validation due to a small sample size and the lack of suitable external datasets aligning with imaging and clinical parameters. Although the importance of external validation was acknowledged, its viability was limited. Future validation efforts are essential to ensure the credibility and generalizability of our model. Analyzing data from multiple institutions would enhance the applicability of the findings. Additionally, the small sample size (n = 15) and fewer tumors in the transitional zone compared to the peripheral zone could potentially confound the results. Manual segmentation of the ROI might introduce inter-observer variability. Future research should prioritize larger, prospective studies involving multiple institutions to validate the potential of radiogenomics, identify relevant imaging biomarkers, and implement them successfully in clinical practice. This study lays the groundwork for standardization, which is critical for these investigations and future research.

## 4. Materials and Methods

### 4.1. Cohort Recruitment

Fifteen men (five from each risk category of low, intermediate, and high, respectively) with clinically localized PCa scheduled for radical prostatectomy (RP) were prospectively recruited into the study. Inclusion criteria were:PSA level of ≤20 ng/mL,T2 disease on imaging and clinically.Patients with a minimum of 10-year expected survival.Ability to give informed consent.

Locally advanced cases and those who could not provide informed consent were excluded from the study. The study had institutional approval (Caldicot number IGTCAL number 5816 and date 6 February 2019). Clinically significant PCa was defined as a lesion with a predicted Gleason’s grade group of 2 or higher, a volume ≥ 0.5 mL, or extra-prostatic extension.

### 4.2. bpMRI Protocol for Imaging Data Acquisition

Pre-biopsy bpMRI scans (T2WI and DWI) were obtained using a 3T scanner. T2WI utilized a turbo spin-echo sequence with a 3.6 mm slice thickness and 0.5 mm in-plane resolution. DWI was acquired through a single-shot echo-planar imaging sequence with 2 mm in-plane and 3.6 mm slice thickness, using diffusion encoding gradients in 3 directions. DWI data with b values of 0, 400, and 1000 s/mm^2^ were used to calculate ADC maps. The images were converted to DICOM format before importing into MATLAB^®^ (Figure 2). Texture features were extracted at a 320 × 320 × 19 resolution, and ROI intensities were normalized to a range of 0 to 1. The DCE sequencing used 3D fast gradient-echo sequences with intravenous Dotarem, a gadolinium-based contrast agent, at a dose of 2 mL/kg, and a temporal resolution of 4 s.

### 4.3. PI-RADS Classification

Two uro-radiologists, each with experience of more than ten years in PCa imaging diagnostics, reviewed the images. The mpMRI, including T2WI, DWI with a corresponding ADC map, and DCE of the index lesion in each patient, were rated using PI-RADS v2.1 on a scale of 1 to 5. Scores from the two readers’ PI-RADS evaluations were used to correlate with the grade of cancer following radical prostatectomy. They also marked on the images a region of interest (ROI) for textural features extraction.

### 4.4. Image Processing and Analysis of Extracted Textural Features

Both T2WI and ADC images were utilized for ROI segmentation, after visually confirmation by experienced radiologists for consistency. ROIs were aligned with histopathological tissue lesions using 3-D printed moulds, as described below. Texture parameters were extracted from segmented T2WI and ADC using MATLAB^®^ software, comprising 22 GLCM and 6 histogram parameters for each ROI. Histograms represent first-order texture features, while GLCMs represent second-order texture features (Appendix A).

### 4.5. Histopathology Protocol

The radical prostatectomy (RP) specimens were labeled, weighed, and fixed in formalin. Patient-specific molds were created from images using 3-D printers. These RP specimens, along with a specialized mold, were used during tissue slicing to ensure accurate alignment with the imaging sections. This precise alignment facilitated the marking of tissue regions of interest (ROIs) corresponding to the MR imaging data. Subsequently, the marked ROI was excised from the tissue area that matched the imaging slice used for extracting radiomic features. This process ensured the macroscopic image co-registration with histologic analyses [21]. Following the tissue slicing and marking process, an experienced uro-pathologist assigned Gleason grades to matched tumor lesions, categorized under the ISUP system [29,30]. The cancer site was then identified on hematoxylin and eosin-stained slides obtained from 10-μm slices of FFPE samples, facilitating microdissection and subsequent genomic analysis.

### 4.6. The OncoScan^®^ FFPE Assay (Genomics Analysis)

There were several challenges to using formalin-fixed paraffin embedded tissue for genetic analysis, due to DNA degradation and natural tumor heterogeneity. These were overcome by utilizing the OncoScan^®^ FFPE Assay Kit. The OncoScan array protocol utilizes SNP probes to assess copy variation and allele frequency, as described in detail in other studies [59].

### 4.7. DNA Extraction Protocol

The QIAamp DNA FFPE Advanced technique, as described in www.qiagen.com (accessed on 22 April 2022) [31,60], efficiently removed paraffin from FFPE tissue blocks without using xylene or trimming excess paraffin. Formalin-induced cross-links in DNA are eliminated through proteinase K digestion and DNA de-crosslinking. Further steps involving uracil-N-glycosylase, RNase A digestion, and DNA binding improve lysis efficiency and yield. Buffers and ethanol are used to remove contaminants, and DNA is eluted in a concentrated form. The amplified DNA is labeled with fluorescent dyes and measured on the ONCOSCAN array, enabling specific sequence binding. The output is converted to a CEL file for analysis, providing valuable genetic information from archived PCa tissues.

### 4.8. Statistical Analysis

The study utilized one-way ANOVA and the Kruskal–Wallis test to determine the variation in radiomic texture features. One-way ANOVA was employed when the data conformed to a normal distribution, while Kruskal–Wallis test was used when the data did not conform to a normal distribution. The significant radiomic texture features (*p*-value ≤ 0.05) were then selected for further analysis.

The genotyping interface was used in the study’s evaluation of the CEL files to confirm the accuracy of the genotyping analysis. Samples having a QC call score of 80% or more were taken into consideration for further analysis whereas samples with a score of less than 80% were excluded from the analysis. The ChAS program was used to import CEL data and create copy numbers from raw intensity to investigate CNV.

The analysis and reporting of copy number and cytogenetics studies, focusing on the alterations in androgen receptor, apoptosis, and hypoxia related genes, was performed using the ChAS program https://www.thermofisher.com/chas (accessed on 8 September 2022). To access the data, we opened the OSCHP files in ChAS and used the Segments tab (rather than the default Karyoview). With an emphasis on copy number gain or loss, ChAS enabled us to visualize and summarize chromosomal abnormalities throughout the genes of interest. This focuses on the overall number of gains and losses across each sample, and attempts to correlate these genetic scores with high, medium, and low Gleason Score categories were made.

Statistical analysis of copy number states in each sample was performed, followed by statistical correlation with radiomics texture features using Pearson’s correlation analysis. SPSS version 22 and R (version 4.2.1; R Core Team, 2022) were utilized for statistical analysis. Model predictions were evaluated using Boosted regression (BR) analysis.

## 5. Conclusions

Radiomic texture features display a significant correlation with genotypes associated with apoptosis, hypoxia, and androgen receptor expression in localized prostate cancer. Integrating these findings with genomics enhances the prediction accuracy not only for prostate cancer grade but also for clinically significant prostate cancer, yielding an impressive AUC of 0.95. However, it is imperative to acknowledge the study’s limitation, as it focuses exclusively on predicting clinically significant prostate cancer within the context of localized disease. Moving forward, additional validation and clinical application are necessary to fully harness the potential of these findings in guiding treatment decisions for localized prostate cancer.

## Figures and Tables

**Figure 1 ijms-25-05379-f001:**
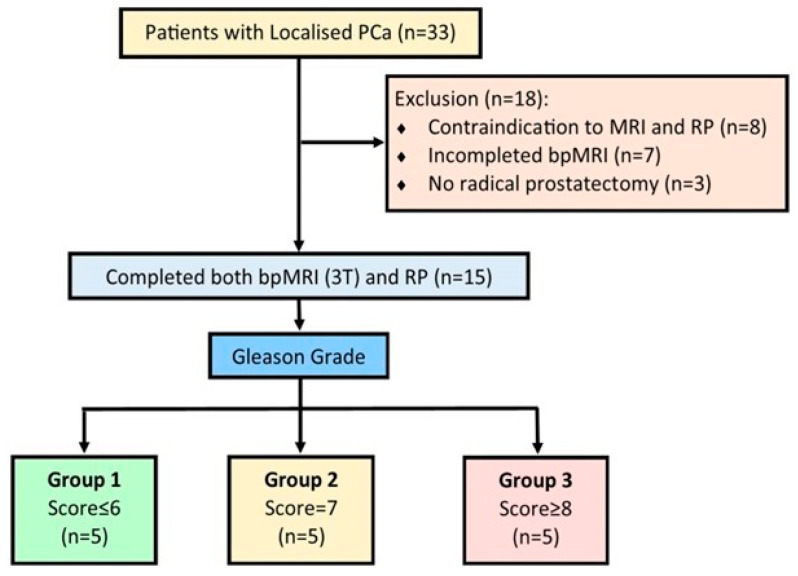
Exclusion and Inclusion criteria.

**Figure 2 ijms-25-05379-f002:**
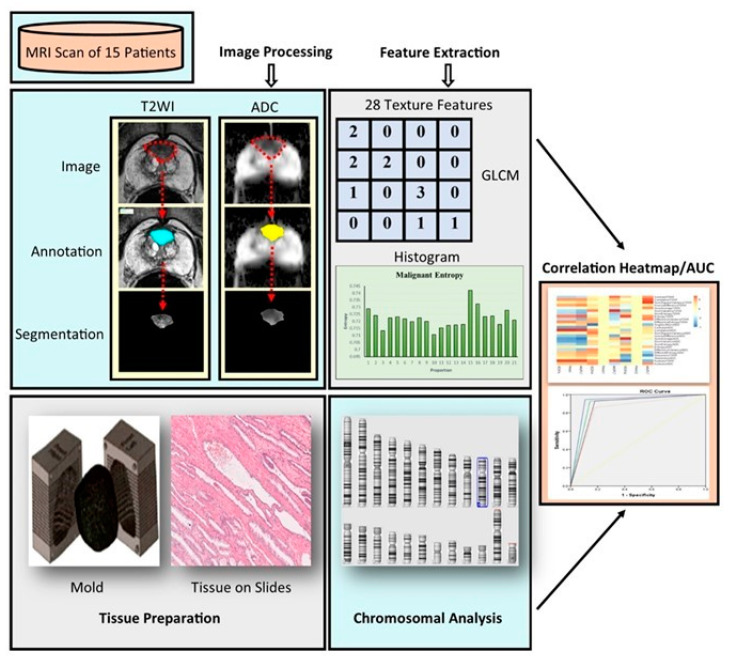
Schematic description of the study. Stage 1 involved the extraction of radiomic textural features from T2WI and ADC images using MATLAB software R2023a. In Stage 2, tissue blocks were marked at ROI corresponding to images and Gleason scored post H and E staining. DNA extraction and ChAS tool analyses chromosome structure abnormalities, such as copy number gains or losses. Pearson’s correlation examines the relationship between radiomics texture features and genetic CNVs. The results are represented in a heatmap graph.

**Figure 3 ijms-25-05379-f003:**
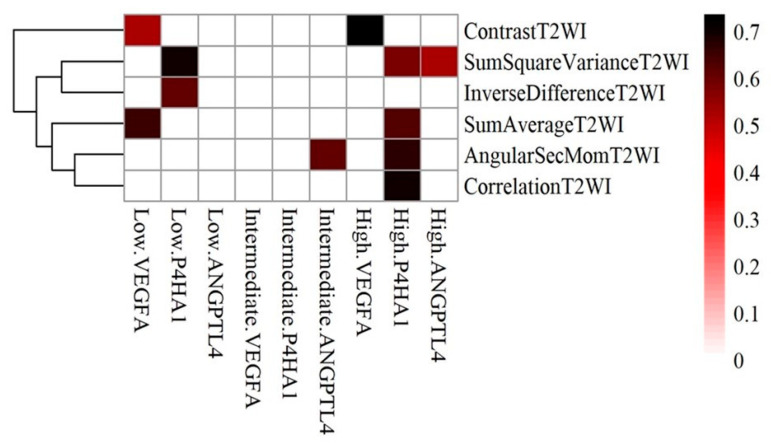
Correlation heatmaps with Hierarchical Clustering showing hypoxia genes against significant radiomic features in relation to Gleason grades of cancer.

**Figure 4 ijms-25-05379-f004:**
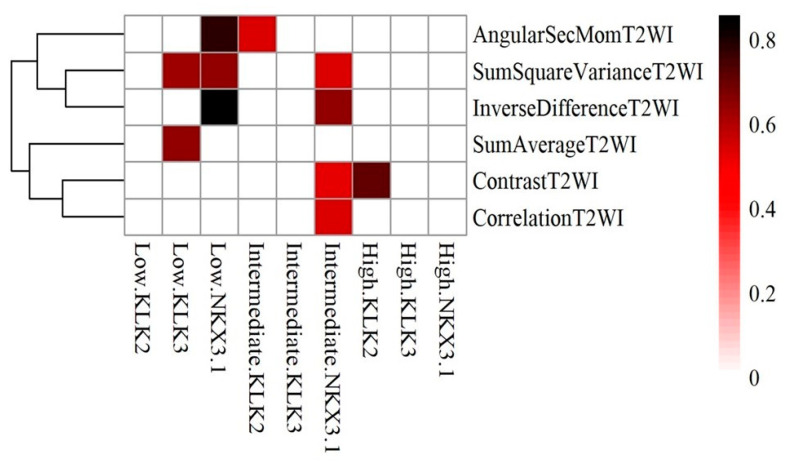
Correlation heatmaps with Hierarchical Clustering of androgen receptor and significant radiomic features in relation to Gleason grade.

**Figure 5 ijms-25-05379-f005:**
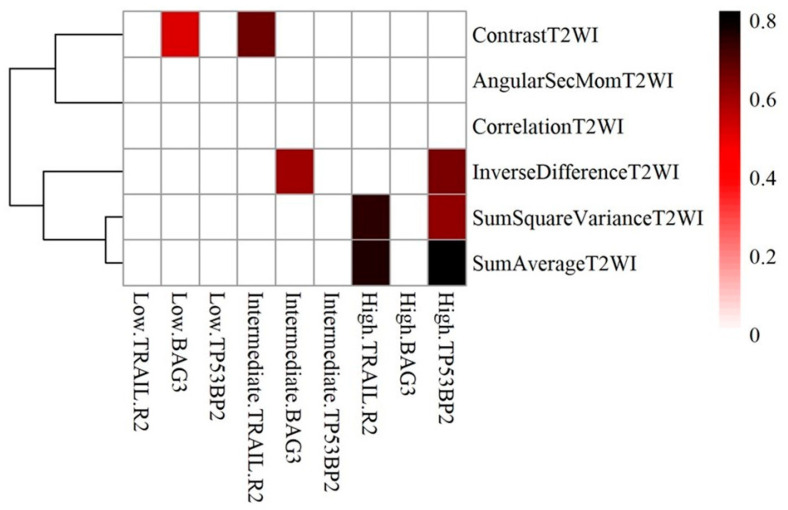
Correlation heatmaps with Hierarchical Clustering of apoptotic and significant radiomic features in relation to Gleason grade.

**Figure 6 ijms-25-05379-f006:**
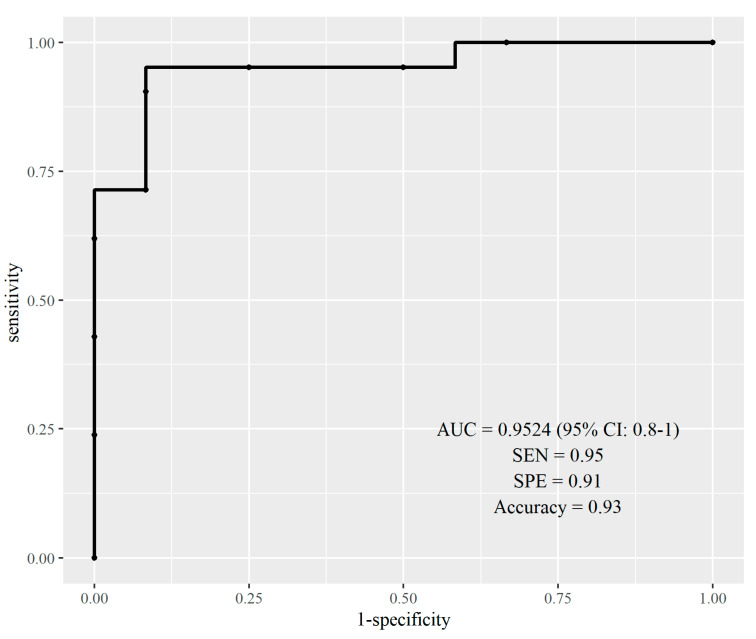
ROC curve of radiogenomics model in predicting clinically significant cancer.

**Table 1 ijms-25-05379-t001:** Demographic data and Gleason grouping.

Characteristics	PCa (n = 15)	(Median-IQR)
Age (years)	15	66–13
Pre-operative PSA (ng/mL)		
PSA 4–20	15	10.8–4.2
Size (mL)	15	51.5–22
Location		
Transitional zone	4	
Peripheral zone	11	
Clinical staging		
cT2a	6	
cT2b	5	
cT2c	4	

**Table 2 ijms-25-05379-t002:** Statistically significant radiomic texture features between the three groups of Gleason scores groups (*p*-value ≤ 0.05).

Radiomic Texture Features	*p*-Value
Contrast T2WI	0.029
Inverse Difference T2WI	0.026
Sum Square Variance T2WI	0.028
Sum Average T2WI	0.029
Difference Variance T2WI	0.032
Angular Secondary Moment T2WI	0.049

## Data Availability

The data is available on request and sharing.

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
