# Peer review of "Radiogenomics Map-Based Molecular and Imaging Phenotypical Characterization in Localised Prostate Cancer Using Pre-Biopsy Biparametric MR Imaging"

_ijms, 2024, doi:10.3390/ijms25105379_

Round 1

Reviewer 1 Report

Comments and Suggestions for Authors

The paper is interesting. Authors constructed a radio genomics model to predict clinically significant PC. Authors correctly reported the limitations, namely the sample size and the single centre. I have some major concerns.

1) A table with definitive pathological examination and pathological/clinical staging should be reported to detail how many patients have clinically significant cancer

2) PC of the transitional and peripheral zone have different criteria to assess PIRADS score. Accordingly, MpMRI reliability for transitional and peripheral cancer differs significantly. Does their model take into account those differences?

3) A prediction model needs external validation beyond greater sample size

4) Authors should correct their conclusion. Since all the patients of the study carries PC the model predicts only clinically significant PC, unless they refer to specific zone (volume) of the prostate. In the latter case, the paper should be deeply revised.

Author Response

Thank you for your comments and suggestions.

  • A table with definitive pathological examination and pathological/clinical staging should be reported to detail how many patients have clinically significant cancer.

ANS: All 15 patients included in this study were diagnosed with localized prostate cancer (<T2 stage clinically) and found to have clinically significant prostate cancer (Gleason score >7), a determination supported by their categorization according to the ISUP (International Society of Urological Pathology) grading system. This system classifies prostate cancer into five grade groups based on the Gleason score, which we regrouped into three categories, as illustrated in Figure 2. Table 2 updated.

  • PC of the transitional and peripheral zone have different criteria to assess PIRADS score. Accordingly, MpMRI reliability for transitional and peripheral cancer differs significantly. Does their model take into account those differences?

ANS: Yes, we're aware that multiparametric MRI (mpMRI)reliability for transitional and peripheral cancer can vary significantly.  The paper was not aimed to answer this particular question as the present study is a proof-of-concept report to show whether it is possible to study and correlate image-based phenotype features of the cancers with the genotypical characters.  This a limitation of the present study worth a focus of research in the future.  There were 4 transitional zone and 11 peripheral zone cancers and the study has proven that radiogenomic-maps can be obtained successfully in both the zones.

  • A prediction model needs external validation beyond greater sample size.

ANS: External validation was unattainable in our study primarily due to two key factors: the limited sample size and the absence of appropriate external datasets matching our imaging and clinical parameters.

Despite these obstacles, we highlighted the critical importance of external validation for future research endeavours. This validation is essential to validate the credibility and generalizability of our model. However, this has been updated in my limitations section of the main manuscript.

  • Authors should correct their conclusion. Since all the patients of the study carries PC the model predicts only clinically significant PC, unless they refer to specific zone (volume) of the prostate. In the latter case, the paper should be deeply revised.

ANS: Has been corrected in the revised manuscript.

Reviewer 2 Report

Comments and Suggestions for Authors

Although the concept radiogenomics has been discussed for several years, clinical application remains limited. This work verified the feasibility of radiogenomics in Prostate Cancer diagnosis, even with a limited cohort number. Minor issues need to be clarified:

1. More detail on the '3D printed Tissue Mold' is needed to clarify how to assure the same orientation of prostate with MRI.

2. All the correlation heatmaps (Fig. 3-5), only used T2WI information. Based on Table 2, ADC also provided significant features. 

3. For the radiogenomics assisted treatment evaluation, MRI radiomics will need to combine with the genomics information from biopsy tissues, more discussion/introduction regarding this is needed.

Author Response

Thank you for your comments and suggestions.

  1. More detail on the '3D printed Tissue Mold' is needed to clarify how to assure the same orientation of prostate with MRI.

ANS: We have updated under Histopathology protocol.  However, this has been clearly referenced in our previous studies

  1. All the correlation heatmaps (Fig. 3-5), only used T2WI information. Based on Table 2, ADC also provided significant features.

ANS: Table 2 has been corrected and the heatmap utilised only statistically significant radiomic texture features between the three groups of Gleason scores groups (p-value ≤0.05).

  1. For the radiogenomics assisted treatment evaluation, MRI radiomics will need to combine with the genomics information from biopsy tissues, more discussion/introduction regarding this is needed. For this study we used radical prostatectomy tissue

ANS: In our study, we addressed the need to integrate MRI radiomics with genomics information obtained from histopathology of radical prostatectomy.  This was done to ensure we use “gold-standard” as reference standard, in particular when it is well-known that there are differences and discrepancies between histopathology of biopsy and radical prostatectomy.  Of course, ongoing future studies will address the point raised by reviewer.

Round 2

Reviewer 1 Report

Comments and Suggestions for Authors

The paper has been amended properly